# Oral Glucose Tolerance Test in Patients with Cystic Fibrosis Compared to the Overweight and Obese: A Different Approach in Understanding the Results

**DOI:** 10.3390/children9040533

**Published:** 2022-04-08

**Authors:** Mirela Mogoi, Liviu Laurentiu Pop, Mihaela Dediu, Ioana Mihaiela Ciuca

**Affiliations:** 1Pediatric Department, University of Medicine and Pharmacy “Victor Babes”, 300041 Timisoara, Romania; pop.liviu@umft.ro (L.L.P.); dediu.mihaela@umft.ro (M.D.); ciuca.ioana@umft.ro (I.M.C.); 2Pediatric Pulmonology Unit, National Cystic Fibrosis Centre, Clinical County Hospital Timisoara, 300226 Timisoara, Romania

**Keywords:** oral glucose tolerance test, cystic fibrosis, obesity, abnormal glucose tolerance, alternative methods

## Abstract

(1) Background: In cystic fibrosis (CF), the oral glucose tolerance test (OGTT) is recommended from 10 years old annually to screen and diagnose cystic fibrosis-related diabetes (CFRD). Alternative OGTT characteristics (glucose curve shape, time to glucose peak, one-hour glucose value, and three-hour glucose value with the new shape curve) were studied in other populations considered at high risk for diabetes; (2) Methods: The study analyses classical and alternative OGGT characteristics from 44 children (22 CF, 22 obese without CF), mean age: 12.9 ± 2.2 years evaluated in a single-center from Romania. (3) Results: In 59.1% of children with CF, the predominant OGTT pattern was: abnormal glucose metabolism or CFRD, with a monophasic curve shape, a late peak glucose level, and 1 h glucose ≥ 155 mg/dL, showing a very different pattern compared with sex and age-matched obese children. Statistical estimation agreement between the late glucose peak (K = 0.60; *p* = 0.005), the 1 h glucose ≥ 155 mg/dL during OGTT (K = 0.69, *p* = 0.001), and the classical method of interpretation was found. (4) Conclusions: Late peak glucose and 1 h glucose level ≥ 155 mg/dL during OGTT can be used for diagnosing the early glucose metabolism alteration in children with CF.

## 1. Introduction

Cystic fibrosis (CF) is a monogenic disease with an estimated incidence of 1 in 2500 newborns [1], which is caused bygenetic mutations in the CFTR gene. In CF, F508del is the most frequently detected mutation [2]. The most commonly diagnosed extra-pulmonary co-morbidity in these children is cystic fibrosis-related diabetes (CFRD) [3]. CFRD does not occur in all patients. It is more prevalent in patients with more severe defects of the cystic fibrosis transmembrane conductance regulator (CFTR) channel (like F508del mutation), those with exocrine pancreatic insufficiency, and the prevalence increases with age [3,4].

Although CFRD has many similarities with Type 2 Diabetes Mellitus (T2DM) and a few with Type 1 Diabetes, it is nevertheless a distinct type of CF with different outcomes and monitoring. The International Society for Pediatric and Adolescent Diabetes (ISPAD), as well as the American Diabetes Association (ADA) guidelines, recommend annual screening for CFRD in children with CF, starting from ten years old [5,6]. The primary recommended screening approach remains the interpretation of fasting glucose (FBG) and the two-hour glucose level (2 h G) during The Oral Glucose Tolerance Test (OGTT). The test results are interpreted following the same criteria used in pediatric practice for screening and diagnosing glucose metabolism alterations and T2DM in overweight and obese children [6,7].

Other OGTT characteristics like glucose curve shape (biphasic, monophasic, or unclassified), time to glucose peak (in early phase or late phase), one-hour glucose value (1 h G), and three-hour glucose value (3 h G) with the new shape curve (the prolonged OGTT) were studied in patients with obesity [8,9,10,11,12]. The monophasic and the unclassified shape curve, the early glucose peak (<30 min), and the value of the 1 h glucose level during OGTT were proved to be associated with lower insulin sensitivity and decreased ꞵ-cell function in childhood obesity, and therefore with a greater risk for developing glucose metabolism alterations, and later diabetes [9,10,11,12]. These characteristics were less studied in other high-risk diabetes populations (e.g., children with CF) [13,14].

This study aims to analyze the classical and alternative OGTT characteristics in children with CF. A secondary aim is to compare the results derived from analyzing OGTT patterns in children with CF with already known patterns found in pediatric patients who are overweight and obese.

## 2. Materials and Methods

### 2.1. Study Design

The paper presents a single-center, retrospective, observational study including data from children with cystic fibrosis who are overweight and obese, evaluated and followed in the IInd Pediatric Clinic and National Cystic Fibrosis Center, Clinical County Hospital Timisoara. The study protocol was developed following the Helsinki Declaration and was approved by the Ethics Committee of Clinical County Hospital (no. 281/21 January 2022). The complete anonymized data sets were used without obtaining individual consent because only pre-existing data were interpreted and all measurements made are part of the local evaluation protocol.

### 2.2. Participants

Children aged 10 to 18 years with CF, evaluated during the annual follow-up visit, were included in the study. The exclusion criteria were: previously diagnosed CFRD; clinical history of pulmonary exacerbation or symptoms of acute infection and steroid treatment at presentation and in the six weeks preceding the evaluation; incomplete data. Genetic data about CF-related genotypes were collected by revising patients’ clinical charts at the National Cystic Fibrosis Center Timisoara.

A group of controls was recruited from children evaluated for excessive weight in the IInd Pediatric Clinic, Clinical County Hospital. These overweight patients were age- and sex-matched with CF patients. Exclusion criteria were: other chronic medical problems or chronic treatment known to be associated with weight gain (e.g., corticosteroids); already diagnosed with impaired fasting glucose, impaired glucose tolerance or diabetes; incomplete data.

### 2.3. Study Variables, Measurements and Definitions

The first phase of the evaluation protocol included the determination of anthropometric measurements: weight (kg), height (m), and body mass index (BMI). The measurement interpretation and z-score calculation were carried out according to World Health Organization (WHO) Reference 2007 criteria [15]: a BMI z-score value above +1 standard deviation (SD) but lower than +2 SD was defined as overweight; obese when BMI z-score was greater or equal to +2 SD; underweight for BMI z-score lower than −2 SD. Pubertal development was evaluated using the Tanner stage score: Tanner stage I was classified as prepubertal; Tanner stage II, III, and IV as pubertal; Tanner stage V as postpubertal.

OGTT’s were performed after a 12 h fast, using 1.75 g/kg (maximum 75 g) of anhydrous glucose. Venous blood samples for glucose measurement were collected at 0, 30, 60, 120, and 180 min. The value of blood glucose level after the first 90 min is not included in the hospital’s laboratory test protocol for OGTT, so it was not evaluated. The OGTT interpretation and classifications were conducted according to the current guidelines [5,6,7]. All 44 children were classified as having one of the following glucose tolerance stages:(a)normal glucose tolerance [NGT: FBG ≤ 126 mg/dL (≤7 mmol/L), 2 h G < 140 mg/dL (<7.7 mmol/L)];(b)impaired glucose tolerance [IGT: 2 h G ≥ 140 (≥7.7 mmol/L) and < 200 mg/dL (<11.1 mmol/L)];(c)impaired fasting glucose [IFG: FBG > 100 mg/dL (5.5 mmol/L) but ≤ 126 mg/dL (≤7 mmol/L), with normal 2 h G and all glucose levels < 200 mg/dL (<11.1 mmol/L)];(d)diabetes [CFRD/T2DM: 2 h G ≥ 200 mg/dL (≥11.1 mmol/L)] with and without fasting hyperglycemia.

In addition, for the CF group, we used a specific definition for modified glucose tolerance known to progress toward IGT and later CFRD [5]: the indeterminate glucose tolerance [INDET: FBG ≤ 126 mg/dL (≤7 mmol/L), 2 h G < 140 mg/dL (<7.7 mmol/L), but 1 h G ≥ 200 mg/dL (≥11.1 mmol/L) during OGTT].

The term “abnormal glucose tolerance” (AGT) was used to define IGT, INDET, and IFG, but not diabetes.

Laboratory assays were performed by the hospital laboratory with reagent from Siemens Healthineers using Dimension RxL_L5 System according to the manufacturer’s protocol (spectrophotometric).

After applying the traditional criteria for glucose level during OGTT, we used three alternative methods based on published definitions [8,13]:glucose curve shape across 2 h and a prolonged OGTT (3 h test): biphasic, monophasic, and unclassified curve. A change in glucose value greater or equal to 4.5 mg/dL (0.25 mmol/L) defines as an “increase” or “decrease”. The monophasic curve is defined by an increase in glucose concentrations followed by a decrease. The biphasic curve is the second rise in glucose concentrations after the initial decrease. The unclassified curve is a gradual increase in plasma glucose levels without a corresponding fall.time of glucose peak ≤ 30 min as early phase, or a late phase when glucose peak > 30 min.1 h G value equal or above 155 mg/dL (8.6 mmol/L).

A blood glucose value below or equal to 70 mg/dL (3.9 mmol/L) was defined as hypoglycemia [5,6].

### 2.4. Statistical Analysis

IBM SPSS v 23.0 and Excel 2007 (Microsoft, Chicago, IL, USA) were used for statistical analysis. The demographic and anthropometric characteristics were calculated as mean, with SD for continuous variables and frequency (%) for categorical ones. The normal data distribution was established with histograms followed by the Shapiro-Wilk test. FBG and glucose level at 30 min during OGTT in the CF group was logarithmically transformed (lg10) to obtain the normal distribution. Differences between groups were assessed using the Student t-test and ANOVA with post-hoc Bonferroni correction. Cohen’s kappa coefficient (K) was used to evaluate the agreement relation between OGTT classical interpretation and alternative methods [16]. The K coefficient was interpreted according to Cohen’s suggestions as follows: values ≤ 0 as indicating no agreement, 0.01–0.20 as poor, 0.21–0.40 as fair, 0.41–0.60 as moderate, 0.61–0.80 as substantial, and 0.81–1.00 as near perfect agreement [17]. For diagnostic accuracy, the sensitivity, specificity, negative predictive value (NPV), and positive predictive value (PPV) were calculated [18]. The confidence interval (CI) was 95% in all cases, and a *p*-value < 0.05 was considered statistically significant.

## 3. Results

### 3.1. Descriptive Data

The study included 44 children: 22 with CF; 5 overweight and 17 obese without CF. The mean age was 12.9 ± 2.2 years old. The control group was selected to match the sex and age of children from the CF group.

Baseline study group characteristics are presented in Table 1. The control group was divided according to BMI z-score into children that are overweight and obese for the statistical analyses.

Because of the selection methods mentioned above, the two study groups did not differ in age, sex, and Tanner stage. As was expected, the differences were statistically significant when we compared the z-scores for weight, height, or BMI (*p* < 0.001) between all groups. Furthermore, it can be observed that the mean glucose value during OGTT reached its highest level at 1 h in the CF group, and the differences were significant compared with the obese group (*p* = 0.003) and the entire control group (including children classified as overweight; *p* = 0.01). In contrast, the non-CF group reached the highest mean glucose level earlier, at 30 min, even if the differences between groups did not reach statistical significance this time. Furthermore, the children with CF tended to have higher means in 2 h G and peak glucose values than the controls, and the differences were significant (*p* < 0.05). Even if the 3 h G mean values were similar in the two groups, we found that the number of children experiencing hypoglycemia at 180 min was more than double in the CF group (5 CF vs. 2 obese). Analyzing these seven cases, it was observed that one child from each group was classified according to classical OGTT interpretation as AGT (INDET from CF group, respective IGT from the control group), and the rest were NGT.

The patients with CF had their genetic testing conducted at diagnosis. The majority, meaning 59.1% of cases, had F508del mutation, 45.5% (n = 10) were homozygous, and 13.6% (n = 3) were in compound heterozygosity form. A small number (n = 2; 9.1%) had a different mutation (G542X) and compound heterozygosity with an unknown allele. In three cases (13.6%), the genetic test results had one unknown mutation, and in four cases (18.2%), we could not find the results. Even if the sample size was not large, the patients are representative of our country and of patients being followed in the Romanian National Center for Cystic Fibrosis. The results were similar with those found in patients for all ages evaluated in our center, where most of the patients had a severe genotype, almost half of them (49.1%) were F508del homozygous, and F508del allele was present in 71.05% of the cases, followed by G542X in a percentage of 6.1% [19].

Pancreatic insufficiency was previously diagnosed in all children with CF, and adequately treated with pancreatic enzyme replacement therapy with no specific gastrointestinal symptoms at the time of the evaluation.

No statistically significant results were found when comparing the OGTT results in children with homozygous F508del to children with other genotypes.

### 3.2. OGGT Patterns by Traditional and Alternative Criteria

The OGTT patterns and the unique combination of traditional and alternative criteria were examined in each group dichotomized by BMI z-score and CF presence or absence, as represented in Table 2.

We found that only a small percent (2.3% of all children: one normal weight child with CF and none from the control group) demonstrated abnormal OGGT patterns by all methods of interpretation. More than half of the children, meaning 54.5% (n = 24; 17 from the control group and seven from the CF group) had the OGTT pattern modified by at least one non-classical method but NGT, according to traditional criteria. None of the children with CF or controls had a normal OGTT pattern by all four methods.

In a more detailed analysis, it was observed that less than a quarter (18.2%) of children met the biphasic curve criteria (one from the CF group and seven without CF) with the majority of them (five from the control group) having a monophasic shape during the classic 2 h test, and the other two after the extended 3 h test. The seven children without CF were classified as NGT but had one other alternative factor for abnormal OGTT. The child with CF, showing a biphasic curve shape during the 2 h test, was diagnosed with CFRD. However, if we exclude the monophasic shape criteria, three children (one from the CF group and two from the obese group) could demonstrate normal OGTT regardless of the method of interpretation used.

The most frequent pattern found in this study in children that are overweight or obese was NGT, with a monophasic shape curve, a peak glucose level ≤ 30 min, and 1 h G < 155 mg/dL. The same pattern was found in five patients with CF, but it was the second pattern type identified in these children. In the CF group, the predominant pattern (59.1%, n = 13) was INDET/IGT/IFG/CFRD with a monophasic curve, peak glucose level after more than 30 min, and 1 h G ≥ 155 mg/dL.

### 3.3. Agreement between OGTT and Alternative Methods

Only a small number of children met the biphasic criteria during the traditional 2 h OGTT (five children with obesity, none of CF group). The authors decided not to include this alternative OGTT characteristic in the agreement analysis. For statistical reasons, we use the term abnormal glucose tolerance (AGT) in the following paragraphs, defining IGT, INDET, and IFG, but not diabetes.

#### 3.3.1. OGTT and Early Peak Glucose (<30 min)

Statistical estimation of agreement between OGTT classical interpretation and early peak glucose showed: no agreement in case of the CF group (K = −0.53 ± 0.19 SD; *p* = 0.004), apoor agreement for the control group (K = 0.01 ± 0.11 SD; but did not reach statistical significance *p* = 0.90), and no agreement for the whole group irrespective of the CF diagnosis (K = −0.40 ± 0.13 SD; *p* = 0.005). The results were similar when the same methods were used to diagnose AGT but not DM. In this study, early peak glucose (<30 min) failed to identify patients with abnormal glucose metabolism in agreement with the classical interpretation of the OGTT.

#### 3.3.2. OGGT and Late Peak Glucose (>30 min)

As it was observed, the majority of children with CF (n = 14; 63.6%) had a late peak glucose level during OGTT. Analyzing the agreement with this OGTT characteristic, it was found that there was a moderate agreement (K = 0.60 ± 0.17 SD; *p* = 0.005) in the CF group, no agreement (K = −0.02 ± 0.17 SD; *p* = 0.90) in children with obesity, and a fair agreement (K = 0.40 ± 0.13 SD; *p* = 0.005) for the study group. The results of agreement were the same for each of the three groups when the late peak glucose was used for diagnosis of AGT (CF group: K = 0.46 ± 0.17 SD, *p* = 0.019; all children: K = 0.31 ± 0.13 SD, *p* = 0.021).

Furthermore, for each situation in which we found an agreement, this alternative method’s Specificity, Sensibility, Positive Predictive Value (PPV), and Negative Predictive Value (NPV) were calculated. The results are presented in Table 3.

#### 3.3.3. OGTT and 1 h Glucose Value ≥ 155 mg/dL

In children with FC, a substantial agreement (K = 0.69 ± 0.16 SD; *p* = 0.001) was found between classical OGTT interpretation and 1 h glucose value ≥ 155 mg/dL during the test. We also found a fair agreement (K = 0.38 ± 0.17 SD, *p* = 0.045) when we aimed to diagnose AGT but not CFRD in the same group. For children with obesity, this alternative method reached only a poor agreement (K = 0.04 ± 0.20 SD) without being statistically significant (*p* = 0.80) for AGT diagnosis. None of the children with obesity were diagnosed with Type 2 Diabetes Mellitus, so in this case the agreement cannot be calculated. For the whole study group, the Kappa coefficient showed a moderate agreement (K = 0.54 ± 0.12 SD; *p* < 0.001) when used for the diagnosis of AGT and diabetes, and a fair agreement (K = 0.35 ± 0.13 SD; *p* = 0.012) in the case of only AGT diagnosis.

The Specificity, Sensibility, PPV, and NPV in each group, where the agreement between these two interpretation methods was found, are presented in Table 4.

### 3.4. Clinical Outcomes and OGTT Classical and Alternative Characteristics

Analyzing the BMI z-score in the CF group, we found that the majority (77.3%, n = 17) were normal-weight children. Only four children (18.2%) met the underweight WHO criteria [15], and one CF patient (4.5%) was obese. At the same time, in the control group, five children (22.7%) were overweight, and 77.3% (n = 17) were obese.

The clinical outcome differences, measured by weight z-score, height z-score, and BMI z-score found in the CF group, are presented in Table 5. The group was dichotomized by traditional OGTT interpretation and two of the alternative significant characteristics (the late peak glucose value and 1 h G ≥ 155 mg/dL during OGTT).

As shown in Table 5, the CFRD group had a better nutritional status evaluated by BMI-z score, weight z-score, and height z-score. Nevertheless, we think that these results are influenced by the very small number of children included in the group (n = 4) and that the only child with obesity and CF is part of this group being diagnosed with CFRD.

## 4. Discussion

As we already know, CFRD is not Type 1 Diabetes, nor T2DM. According to ADA, this type falls under the diabetes category of diseases of the exocrine pancreas (Type 3c) [6]. The development of this CF-related complication is insidious, with symptoms strongly related to the clinical course of the disease, a negative impact on pulmonary function and nutritional status, and consequently on mortality [20,21,22,23]. The prevalence of CFRD varies between studies and is positively associated with screening rates [21]. In a recently published European study, the prevalence of CFRD was 0.8% in patients younger than 10 years old, and 9.7% in the 10 to 19 years old group [24]. In our group, the prevalence was higher (18.2%) and was similar to other countries like the Czech Republic (19.2%), Slovenia (20%), or Serbia (15.6%) [24]. Furthermore, it is demonstrated that CFRD is associated with severe genotypes, pancreatic insufficiency, and female gender [4,21,24,25], characteristics that were predominant in our group of children.

At present, there is a consensus regarding the critical need for early diagnosis of glucose metabolism abnormalities in patients with CF. The OGTT remains the recommended test for screening glucose metabolism abnormalities in these children [5,6]. Although this test has been used for over 100 years, it still has multiple limitations such as the need for fasting time, multiple timed venous sampling, poor reproducibility of the results, lack of specific criteria, and cut-offs in different groups [8,26]. The OGTT’s interpretation “gold standard” criteria remain the FBG and the 2 h G values derived from T2DM diagnosis guidelines [5,6]. In a study that included 1128 patients with CF, the variability of 2 h blood glucose was 1.5 to 1.8-fold higher than the general population [27]. In order to overcome these issues and for a better understanding of the results, alternative OGTT characteristics were proposed. These characteristics were studied in other at-risk for diabetes populations, but less studied in children with CF [8,9,10,11,12,13,14].

In a systematic review about diabetes and prediabetes in children with CF, Mozzilo et al. underline that prediabetes was one of the most relevant predictors for deterioration of lung function, and has a significant impact on the growth and nutritional status of the patients [20]. In another study, peak blood glucose ≥ 8.2 mmol/L (≥147 mg/dL) during OGTT was associated with declining weight z-score and lung function in the preceding 12 months [23]. In this paper, the impact of glucose metabolism on respiratory function has not been evaluated, only the growth and weight. The mean weight was the lowest in the CF study group classified as NGT, but the differences between groups were not statistically significant (*p* = 0.15). We think these somewhat discordant results were because the NGT group and the CFRD group are relatively small groups (n = 8, respective n = 4), and those groups included the children with extreme weights. The way of interpreting the OGTT results could have been another factor. For example, one of two underweight children classified as NGT had a peak glucose level at 150 mg/dL, and the other had severe hypoglycemia (<3 mmol/L) at the 3 h determination. The literature data suggested that hypoglycemia during OGTT is an early clinical manifestation of T2DM [28]. Therefore, it is possible that these patients, depending on the used criteria [23,28], can be included in other classes of abnormal glucose metabolism. Finally, a pertinent conclusion cannot be made as the results presented are influenced by the small number of children evaluated and the lack of longitudinal anthropometric data. In such situations, it would be better to look at the individual growth charts and assess the growth velocity and weight oscillations rather than the absolute value of the z-scores.

In this study, more than half of the investigated children with CF already had different stages of glucose metabolism alteration (63.3% by classical criteria vs. 77.4% using alternative criteria) irrespective of their nutritional status. None of the evaluated patients with CF could demonstrate a “normal” OGTT by traditional and alternative criteria. The results were similar to the ones already published [13]. Furthermore, regular control of glucose metabolism is needed in these children. There are a few data regarding the adequate time frame for repeating OGTT in patients diagnosed with AGT but not CFRD.

It is important to look closer at the OGTT results, and not only at the 2 h G value. The use of new diabetes technologies (the continuous glucose monitoring systems—CGMS) in patients with CF highlighted abnormal postprandial glucose elevations [29]. The glucose peaks on CGMS correlated with OGTT intermediate values (at 30, 60, and 90 min), but not with the 120 min glucose values [30].

In this study, the most frequent OGTT pattern found in children with CF was: a monophasic shape curve, late peak glucose (>30 min), and a 1 h G ≥ 155 mg/dL. The monophasic shape is the predominant shape in individuals’ NGT without CF, but high false-positive rates were found in individuals already diagnosed with prediabetes [8]. The biphasic pattern has demonstrated a lower risk-profiling for T2DM in obese pediatric patients [31]. Only a few studies evaluated this OGTT characteristic in children with CF, and the results were similar to ours. In another study, only 7.6% of children demonstrated a biphasic shape curve, and the curve shape characteristic did not identify patients with early ꞵ cell dysfunction [13]. However, we found one child (4.5%) displayingthis pattern, and he was diagnosed with CFRD. The late peak glucose (>30 min) was present in more than half of the children with CF (63.3%). This pattern seems to be characteristic for patients with CF [13,28], and different from the healthy and obese population characterized by an early glucose peak [8,28,32]. The best results were obtained when using the late peak characteristic for diagnosis of AGT, including CFRD in children with CF (K = 0.60; *p* = 0.005) with good sensitivity (85.7%) and specificity (75%). The 1 h G value ≥ 155 mg/dL criteria, was also present in a significant number of the evaluated CF children (68.2%). The statistical agreement with the classical OGTT characteristics achieved statistical significance when used to diagnose AGT, including CFRD in the CF group (K = 0.69; *p* = 0.001) with a sensitivity of 92.9% and a specificity of 75%. This blood glucose determination, named the mid-OGTT glucose value, has different cut-off values (155 mg/dL (8.6 mmol/L) in this study and Tommerdahl et al. [13], and a lower value (140 mg/dL) in others [33,34]). Some authors proposed a mid-OGTT value greater than 140 mg/dL to be considered a new distinct glucose tolerance class from thefourpre-established tolerance stages in patients with CF [33,34]. Piona et al. showed that the patients with 1 h G between 140 and 199 mg/dL already have a significant impairment in insulin sensitivity and altered beta cell function [33]. In another CF study focused on this particular OGTT characteristic a 4-fold higher risk of CFRD was reported in a patient with 1 h G value above 160 mg/dL [35]. More than ten years ago, Brodsky et al. showed that plasma glucose elevations at non-classical OGTT times are common in CF, and an increasing value of 1 h G above 140 mg/dL is associated with worse pulmonary function in these patients [34].

There are some important limitations to consider when interpreting the results: (1) the small number of children evaluated, representing the patients followed-up in a single center; (2) did not evaluate the relationship between OGTT patterns and the pulmonary function; (3) the clinical outcomes were retrospectively and cross-sectional collected; (4) more extensive longitudinal and multicenter studies are needed; (5) the patients were evaluated during a period of stable clinical condition.

## 5. Conclusions

This study is the first that analyses the use of alternative OGGT criteria, beyond the traditional ones, in children with CF, compared to their peers that are overweight or diagnosed with obesity. The control group was chosen because the classical criteria for the OGTT interpretation and the cut-off values are derived from the criteria used in children at risk for developing T2DM. Overweight and obese children are the most important part of this aforementioned group.

In recent years, with the increase in published data, new implemented therapies, new forms of insulin, and increasing access to glucose monitoring systems, the need for specific CF criteria and cut-off value for the early diagnosis of glucose metabolism alteration has emerged.

In summary, in children with CF, other OGTT characteristics like late peak glucose (after more than 30 min) and 1 h G greater or equal to 155 mg/dL, can be used to diagnose abnormal glucose metabolism. The monophasic shape is the predominant curve shape found in this age group of CF children with or without CFRD. Children with CF demonstrated a different glucose pattern during OGTT than obese children. CFRD has only a few common characteristics with T2DM. Thus, the diagnosis criteria and glucose tolerance stages need to be more specific.

Despite mentioned limitations, the OGTT is still the ‘gold standard’ for CFRD diagnosis. The majority of CF centers declared a screening rate below 50% [36]. For this reason, some authors tried to find alternative methods to shorten the period needed to do the test, or to use the lower cut-off value for glycated hemoglobin level (HbA1c > 5.4%) as an intermediate step approach in selecting the patients for OGTT [37]. However, the present literature data are still scarce. Further evaluation of these screening and diagnostic methods is needed.

When we look at the OGTT results, if we know that there is more at play than the level of FBG and the 2 h G value, it is possible to gain new reasons for conducting repeated tests, and to easily convince more parents and their children of their importance. The new information gathered from this relatively accessible evaluation can be used in a more individualized treatment approach. For example, the late peak glucose characteristic can be used to tailor other treatment choices (e.g., type of insulin used, time of administration, meal planning).

In this paper are presented the preliminary results from the first phase of a prospective study aimed to develop a new local early diagnosis, follow-up, and treatment protocol for glucose metabolism alterations in children with CF, based on risk stratification evaluated by classical OGTT criteria and alternative methods. Patients with a high risk of developing CFRD may benefit from increased monitoring, dietary change, intermittent insulin treatment during illness, steroid treatment, and finally, better outcomes in managing this challenging disease.

## Figures and Tables

**Table 1 children-09-00533-t001:** Baseline clinical and laboratory data in children with and without cystic fibrosis.

Characteristics	Children with CF	Control Group			*p*-Value
		Overweight	Obese	Total	
Number (n)	22	5	17	22	-
Age, years	13.1 ± 2.2	12.8 ± 2.5	12.7 ± 2.2	12.8 ± 2.2	0.69
Female, n (%)	17 (77.3)	4 (80.0)	13 (76.5)	17 (77.3)	-
Tanner stage, n (%)					
Prepubertal	5 (22.7)	0 (0.0)	3 (17.6)	3 (13.6)	-
Pubertal	11 (50.0)	3 (60.0)	8 (47.1)	11 (50.0)	-
Postpubertal	6 (27.3)	2 (40.0)	6 (35.3)	8 (36.4)	-
Weight z-score	−1.11 ±1.10	1.65 ± 0.17	2.30 ± 0.35	2.15 ± 0.42	***<0.001*** ^a,b^
Height z-score	−0.83 ± 1.47	0.27 ± 0.20	0.44 ± 1.00	0.40 ± 0.88	***0.002*** ^b^
BMI z-score	−0.90 ± 1.26	1.76 ± 0.14	2.24 ± 0.16	2.13 ± 0.25	***<0.001*** ^a,b^
OGTT					
FBG (mg/dL)	86.8 ± 1.04	92.6 ± 10.21	85.2 ± 10.19	86.9 ± 10.43	0.78
30 min glucose (mg/dL)	161.1 ± 1.05	173.6 ± 28.51	145.9 ± 25.11	152.2 ± 27.85	0.36
1 h glucose (mg/dL)	175.5 ± 42.36	168.4 ± 57.67	136.1 ± 31.14	143.5 ± 39.54	***0.01*** ^b^
2 h glucose (mg/dL)	144.7 ± 57.07	120.8 ± 9.68	116.4 ± 18.15	117.4 ± 16.50	** *0.03* **
3 h glucose (mg/dL)	105.0 ± 40.81	105.6 ± 16.04	97.4 ± 23.16	99.2 ± 21.68	0.56
Peak glucose (mg/dL)	191.5 ± 40.30	197.4 ± 41.70	155.4 ± 24.78	165.0 ± 33.50	***0.02*** ^b^

CF—cystic fibrosis; BMI—body mass index; OGTT– oral glucose tolerance test; FBG—fasting blood glucose. Significant differences for the pairwise comparisons are indicated by the letters ‘^a^’ or ‘^b^’. Comparisons: ^a^—children with CF vs. overweight children without CF; ^b^—children with CF vs. obese children without CF. percentages are calculated by column.

**Table 2 children-09-00533-t002:** The frequencies of glucose metabolism alteration according to classical and alternative OGTT criteria in participants with and without FC dichotomized by BMI z-score.

		Children with CF				Control Group Children			Total n (%)
	Nutritional Status	Under Weight	NW	Obese	Total n (%)	Over Weight	Obese	Total n (%)	
	Number, (n)	4	17	1	22	5	17	22	44
Classical OGTT Criteria:									
Normal		2	6	-	8 (36.4)	4	15	19 (86.4)	27 (61.4)
IGT/INDET/IFG		-	10	-	10 (45.4)	1	2	3 (13.6)	13 (29.5)
Diabetes		2	1	1	4 (18.2)	-	-	-	4 (9.1)
**Patterns of OGTT—traditional and alternative citeria**									
NGT; monophasic, peak glucose ≤ 30 min, 1 h G < 155 mg/dL		2	3	-	5 (22.7)	1	6	7 (31.8)	12 (27.3)
NGT; monophasic, peak glucose ≤ 30 min, 1 h G ≥ 155 mg/dL		-	1	-	1 (4.5)	-	-	-	1 (2.3)
NGT; monophasic, peak glucose > 30 min, 1 h G < 155 mg/dL		-	1	-	1 (4.5)	-	2	2 (9.2)	3 (6.8)
NGT; monophasic, peak glucose > 30 min, 1 h G ≥ 155 mg/dL		-	1	-	1 (4.5)	1	2	3 (13.6)	4 (9.1)
NGT; biphasic, peak glucose ≤ 30 min, 1 h G < 155 mg/dL		-	-	-	-	1	4	5 (22.7)	5 (11.4)
NGT; biphasic, peak glucose ≤ 30 min, 1 h G ≥ 155 mg/dL		-	-	-	-	1	-	1 (4.5)	1 (2.3)
NGT; biphasic, peak glucose > 30 min, 1 h G ≥ 155 mg/dL		-	-	-	-	-	1	1 (4.5)	1 (2.3)
IGT/INDET/ IFG;monophasic, peak glucose ≤ 30 min, 1 h G < 155 mg/dL		-	-	-	-	1	1	2 (9.2)	2 (4.5)
IGT/INDET/IFG; monophasic, peak glucose ≤ 30 min, 1 h G ≥ 155 mg/dL		-	1	-	1 (4.5)	-	-	-	1 (2.3)
IGT/INDET/IFG; monophasic, peak glucose > 30 min, 1 h G < 155 mg/dL		-	1	-	1 (4.5)	-	-	-	1 (2.3)
IGT/INDET/IFG; monophasic, peak glucose > 30 min, 1 h G ≥ 155 mg/dL		-	8	-	8 (36.7)	-	1	1 (4.5)	9 (20.4)
CFRD/T2DM; monophasic, peak glucose > 30 min, 1 h G ≥ 155 mg/dL		2	-	1	3 (13.6)	-	-	-	3 (6.8)
CFRD/T2DM; biphasic, peak glucose ≤ 30 min, 1 h G ≥ 155 mg/dL		-	1	-	1 (4.5)	-	-	-	1 (2.3)

NW–normal weight; CF—cystic fibrosis; OGTT—oral glucose tolerance test; NGT—normal glucose tolerance; IGT—impaired glucose tolerance; INDET—indeterminate glucose tolerance; IFG—impaired fasting glucose; CFRD—cystic fibrosis-related diabetes; T2DM—type 2 diabetes mellitus; 1 h G-1 h glucose level during OGTT.

**Table 3 children-09-00533-t003:** Sensitivity and Specificity of alternative OGTT interpretation method (delayed peak glucose, >30 min) used for diagnosis of abnormal glucose tolerance in a group of children.

	Sensitivity 95% CI	Specificity 95% CI	PPV 95% CI	NPV 95% CI
** *AGT (including CFRD):* **				
Cystic Fibrosis group	85.7%	75.0%	85.7%	75.0%
All children	76.5%	66.7%	59.1%	81.8%
** *AGT (without diabetes):* **				
Cystic fibrosis group	90.0%	58.3%	64.3%	87.5%
All children	76.9%	61.3%	45.5%	86.4%

OGTT—oral glucose tolerance test; AGT—abnormal glucose tolerance; CFRD—cystic fibrosis-related diabetes; PPV—positive predictive value; NPV—negative predictive value.

**Table 4 children-09-00533-t004:** Sensitivity and Specificity of alternative OGTT interpretation method (1 h glucose value ≥ 155 mg/dL) used for diagnosis of abnormal glucose tolerance in a group of children.

	Sensitivity 95% CI	Specificity 95% CI	PPV 95% CI	NPV 95% CI
** *AGT (including CFRD):* **				
Cystic Fibrosis group	92.9%	75.0%	86.7%	85.7%
All children	82.5%	74.1%	66.7%	87.0%
** *AGT (without diabetes):* **				
Cystic fibrosis group	90%	50.0%	60.0%	85.7%
All children	76.9%	61.3%	45.5%	86.4%

OGTT—oral glucose tolerance test; AGT—abnormal glucose tolerance; CFRD—cystic fibrosis-related diabetes; PPV—positive predictive value; NPV—negative predictive value.

**Table 5 children-09-00533-t005:** Clinical outcomes in children with cystic fibrosis categorized by OGTT characteristics.

Clinical Characteristics	Classical OGTT			Alternative OGTT		*p*-Value
	NGT (n = 8)	AGT (n = 10)	CFRD (n = 4)	Late Peak Glucose (n = 14)	1 h G ≥ 155 mg/dL (n = 15)	
*Weight z-score*	−1.76 ± 0.89	−0.92 ± 0.63	−0.26 ± 1.81	−0.74 ± 1.04	−0.89 ± 1.14	0.15
*Height z-score*	−1.30 ± 1.78	−0.90 ± 1.23	+0.30 ± 0.88	−0.41 ± 1.31	−0.69 ± 1.30	0.34
*BMI z-score*	−1.43 ± 1.18	−0.52 ± 0.76	−0.79 ± 2.24	−0.72 ± 1.29	−0.69 ± 1.24	0.63

BMI—body max index; OGTT—oral glucose tolerance test; NGT—normal glucose tolerance; AGT—abnormal glucose tolerance; CFRD—cystic fibrosis-related diabetes.

## Data Availability

Data are contained within the article, the raw data are not publicly available due to reasons of privacy.

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
