# Peer review of "Oral Glucose Tolerance Test in Patients with Cystic Fibrosis Compared to the Overweight and Obese: A Different Approach in Understanding the Results"

_children, 2022, doi:10.3390/children9040533_

Round 1

Reviewer 1 Report

The study by Mogoi and colleagues nicely supports the emerging opinion that early glucose monitoring is required in children even in the absence of overt diabetes and will be of particular interest to clinicians working in the field of CF.

I have only minor comments:

  1. Overall, the paper is well presented and written. However, the language is a little casual in places (beginning sentences with 'So') and would benefit from another round of editing. In particular, please refrain from the use of 'CF children' and 'obese ones' and instead use the preferred terminology of 'people/children with CF/obesity'
  2. Lines 162-163: Please clarify what is meany by control group here. Do you mean children who are overweight? The obese group has already been specifically mentioned.
  3. It would be very useful for the non expert reader to have a brief description of the significance of the alternative measures i.e. what does a monophasic and biphasic curve represent?
  4. Does genotype play a role? Is glucose metabolism worse in those homozygous for F508del? Even if the data were split between those homozygous for F508del and all other genotypes, this would be very helpful. If there is no difference, please state this in the results section.

Reviewer 2 Report

This is a very interesting study on an important topic. 

A few minor commets.

3.1 Descriptive data. I would be interested to know if the sample was represectative, i.e. What are the corresping percentages of the mutations in the Romanian CF population? 

There is an interesting article by Brodsky et al on Diabetes Care 34;292-295, 2011 that could be mentioned to stress the importance of your findings as elevation of 1 hour glucose on OGTT was associated with worse pulmonary function.

Paragraphs 3.3.1 and 3.3.3 are a little confusing I think. Maybe you could rephrase slight agreement.

At the discussion part when you discuss the different cut off values used you could mention the guidelines that you used once again. 

This is a single center study, I think that this is a limitation that could be added as we are not sure how genralizble these results are.
